# Time to initiate trophic feeding and predictors among preterm neonates admitted at General Hospitals in Tigray, 2025

Teklebrhan Welderufael Kidane[1¤a]*, Zeray Baraki[1¤a], Asefa Iyasu[1¤a], Tekle Gebremeskel Ygzaw[2¤a], Binyam Gebrehiwet Tesfay[3¤b], Ngsti Gebremichael Beyene[1¤a], Teklewoini Mariye Zemicheal[1¤a], Nebiat Desale Gidey[2¤a], Geberziher Gebreslassie Welearegay[2¤a]

**1** Axum University College of Health Sciences, Axum, Ethiopia, **2** Araya Kahsu College of Health Sciences, Aksum, Ethiopia, **3** Adigrat University College of Health Sciences, Adigrat, Ethiopia

¤a current address: Axum Northern Ethiopia
¤b current address: Adigrat Northern Ethiopia
* welderufaelteklebrhan@gmail.com

## Abstarct

### Back ground

Trophic feeding generally refers to providing small quantities of enteral feeding soon after birth. A study in Ethiopia highlighted substantial delays in starting therapeutic feeding (TF) for newborns. While guidelines recommend initiating TF within 24 hours of birth, an alarming 80–90% of infants did not begin feeding within 48 hours. Only 20% started TF within the advised timeframe. Furthermore, 29% of these infants did not survive until discharge, and 86.2% experienced extrauterine growth restriction. As a result, this study aims to assess the time to initiation of TF and its predictors among preterm neonates in the Tigray region.

### Method

A prospective, institutional-based follow-up study was conducted on 193 preterm neonates admitted to the Neonatal Intensive Care Unit, with participants selected using systematic random sampling from a group of public hospitals.:The data collection period was from December 20, 2024, to February 30, 2025. Data was entered into Epi-Data version 4.7 and then exported to STATA version 14 for cleaning and analysis. To compare survival curves, Kaplan-Meier analysis and the log-rank test were used, bivariate and multivariate Cox regression analysis were used, all statistical tests were considered significant at a p-value of <0.05.

### Result

A total of 193 neonates were followed for 8382 person-hours of risk time and 173 (89.6%) of neonates were initiated trophic feeding. The incidence rate of initiating

**Data availability statement:** All relevant data are within the paper and its Supporting Information file.

**Funding:** The author(s) received no specific funding for this work.

**Competing interests:** The authors have declared that no competing interests exist.

**Abbreviations:** APGAR: Appearance, Pulse, Grammies, Activity, Respiration, BW: Birth weight, TF: Trophic Feeding, GA: Gestational Age, KMC: Kangaroo Mother Care, NEC: Necrotized Enterocolitis, NICU: Neonatal Intensive Care Unit, PNA: Perinatal Asphyxia, RDS: Respiratory Distress Syndrome, TF: Trophic feeding.

trophic feeding was 2 per 100 person hours' observations with a median time of 45 hours (95% CI: 42−56). Birth weight <1500 gram (AHR: 0.16,95% CI:0.075–0.35), APGAR score at first minute<7 (AHR: 0.46,95% CI:0.26–0.76),APGAR score at fifth minute<7 (AHR: 0.38,95%CI:0.21–0.68, having respiratory distress syndrome (AHR: 0.41,95% CI:0.25–0.66, and absence of Kangaroo mother care (AHR: 0.41,95% CI:0.21–0.77), were statistically Significant associated factors for the delay of initiation of trophic feeding.

## Conclusion

In this study, a significant delay in the initiation time of trophic feeding. Therefore, health institutions should work on very low birth weight, APGAR scores below seven at one and five minutes, the presence of RDS, and the absence of KMC to shorten the initiation time and reduce complications associated with the delay.

## Introduction

Trophic feeding (TF) involves giving preterm infants small amounts of enteral feeding (EF) shortly after birth typically around 1–2 mL/kg per dose or 10–15 mL/kg per day and this practice is designed to nourish and stimulate the developing gastrointestinal tract while providing minimal enteral nutrition which plays an essential role in promoting intestinal growth improving feeding tolerance preparing the gut for more substantial feedings and supplying low-calorie nourishment that supports the fragile digestive systems of these vulnerable infants [1,2].

Preterm birth is characterized as delivery occurring before 37 full weeks of gestation, or less than 259 days from the first day of a woman's last menstrual cycle [3,4]. These infants experience a nutritional emergency as they are suddenly removed from nutrient rich environment in-utero to the extra uterine life where nutrition is harder to initiate and later on maintain [5].

During the neonatal period, appropriate nutritional support is essential to reduce the chances of both immediate and lasting adverse effects [6]. It guarantees that the infant obtains the essential nutrients needed to align with the growth rate and body composition typical of a healthy fetus at the same gestational age (GA) and this encompasses factors like weight, length, head circumference, organ dimensions, tissue makeup (including cell quantity and structure), as well as nutrient levels in blood and tissues, along with overall developmental advancement [7].

The World Health Organization (WHO) 2021 strongly recommends, based on moderate-certainty evidence, that enteral feeding through a nasogastric tube should be started as early as possible, ideally within the first day after birth, for preterm or low birth weight (LBW) infants, including those weighing less than 1.5 kg or born before 32 weeks of gestation.This recommendation is supported by evidence showing moderate benefits, including reduced mortality and shorter hospital stays [8].

The two classifications with recommendations of TF feeding practice according to the Ethiopian Neonatal Intensive Care Unit (NICU) guideline 2021 in preterm

neonates are early initiation of TF, starting within 24 hours of birth, and delayed initiation of TF, starting after 24 hours of birth [9].

Late initiation of TF is a worldwide issue in NICUs and has been recognized as a major independent factor influencing poor growth in preterm neonates [10]. Globally, complications arising from preterm birth are the leading cause of neonatal deaths accounting for 35% of the 3.1 million neonatal deaths each year with 15 million preterm births globally, those born before 32 weeks of gestation are at the highest risk of illness and death [11]. Similarly in Ethiopia, preterm birth complications continue to be a major cause of neonatal mortality and despite global efforts to reduce these complications, the rates of preterm birth remain high with significant delays in interventions such as early trophic feeding [11–13].

Some of the inherent that delay to initiate TF were Increased incidence of necrotizing enterocolitis NEC, prolonged hospital stays, and higher mortality rates [10,14].

TF is a core and widely acknowledged approach that aids in reducing the risk of complications linked to enteral fasting, including necrotizing enterocolitis (NEC), infections, prolonged hospital stay, metabolic problems, and postnatal growth failure (PGF) in preterm neonates [14–21].

Delayed start of TF in preterm neonates can lead to inadequate growth, heightening the chances of nutritional deficiencies, hindered brain development, sepsis, feeding difficulties, low weight, and reduced survival rates among preterm neonates [10,13,22,23]. Similarly, late TF and Prolonged hospitalizations for preterm infants result in a considerable financial strain on families and the healthcare system, stressful hospital stay, costs associated with neonatal intensive care, continued healthcare needs, and educational requirements [24,25].

Preterm birth is the primary cause of neonatal and under-5 mortality worldwide, and the associated healthcare burdens and nutrition-related complications are unsustainable, especially in resource-constrained areas [26]. Every year, there are 15 million preterm births globally, with infants born before 32 weeks of gestation facing the greatest risk of illness and mortality from this undernutrition in these preterm infants is linked to severe outcomes, including a higher risk of death [11].

International studies have found that 80–90% of these newborns did not begin TF until they were 48 hours old which indicates a notable delay in the commencement of TF for preterm infants [14]. In Africa, a limited number of infants receive early minimal feeding, leading to serious repercussion [27]. For instance, a study conducted in Uganda found that starting TF more than 48 hours after birth was linked to postnatal growth failure compared to initiating feeding sooner [5]. In addition study in Zambia reviled that neonate delayed enteral feeding cause 66% death compared with 6% in the early feeders [28].

Research conducted in Ethiopia revealed significant delays in transitioning to TF for neonates, as guidelines suggest it should begin within 24 hours of birth; however, alarmingly, 80–90% of infants did not start feeding within 48 hours, and only 20% received TF within the recommended time frame, while additionally, 29% of these infants did not survive until discharge, and 86.2% faced extra uterine growth restriction [10,29,30].

Several factors were recognized as major predictors of this delay, including an APGAR score below 7 at one minute, APGAR score less than seven at five minute, GA under 34 weeks, respiratory distress syndrome (RDS), Very low birth weight, hemodynamic instability, perinatal asphyxia (PNA), cesarean section (CS) delivery, Absence of KMC service,sepsis, hypothermia and being born out of the study hospitals [10,13,14,29].

The timing of TF in preterm neonates is critical for their health, yet significant gaps exist in understanding this process, especially in general hospitals in Tigray, Ethiopia. Current literature demonstrates limited evidence on the timely initiation of TF among preterm neonates admitted to Neonatal Intensive Care Units (NICUs), and many studies do not utilize a prospective follow-up methodology. This lack of localized data contributes to delays in TF initiation, potentially impacting health outcomes for these vulnerable infants. Identifying specific barriers that affect the timing of TF, such as cultural attitudes, resource constraints, and the training needs of healthcare providers, is essential for informing improvements in clinical practices.

Despite recognizing the benefits of early TF, healthcare providers may still hesitate to start feeding promptly due to concerns about the stability of the infant, inadequate training in nutritional guidelines, and systemic barriers within the NICU setting. This study aims to investigate the predictors of timely TF initiation among preterm neonates in Tigray and will contribute valuable insights to the existing evidence base. By elucidating the factors that delay TF, the research can provide targeted recommendations to enhance care quality, reduce healthcare costs, and improve long-term health outcomes for families, thereby ensuring that preterm infants receive the critical nutrition they need in a timely manner on time.

## Methods and materials

### Study area and period

The study was conducted in six general hospitals in Tigray, Ethiopia, from December 20, 2024, to February 30, 2025. The region comprises two specialized hospitals, 14 general hospitals, 24 primary hospitals, 230 health centers, and 741 health posts. All general hospitals are organized into different service areas, including term and preterm care, isolation, procedure rooms, as well as KMC, and maternal waiting rooms. Each Neonatal Intensive Care Unit (NICU) ward has approximately 10–16 healthcare providers. The two specialized hospitals in the region each have separate admission rooms for KMC services and employ 20–40 healthcare workers.

### Study design

Institution-based, prospective follow-up study design was conducted.

### Source of population

All preterm neonates (< 37 weeks GA) admitted to NICU during the study period at public general hospitals in, Tigray, Ethiopia.

### Study population

All preterm neonates (< 37 weeks GA) admitted to NICU during the study period at the selected public general hospitals.

### Eligibility criteria

**Inclusion criteria.** Preterm neonates (less than 37 weeks GA) admitted to the neonatal intensive care unit (NICU) of the study general hospitals during the study period.

**Exclusion criteria.** Preterm contraindicated for feeding like pre-diagnosed Stage II or III necrotizing enterocolitis and Stage III asphyxia. Neonates with significant gastrointestinal malformations, such as esophageal atresia, imperforate anus, duodenal and jejunal atresia, intestinal obstruction or perforation, and paralytic ileus.

Neonates with, unknown APGAR scores, gestational ages and home deliveries were excluded from the study.

### Sample size determination

Sample size was calculated using Stata version 14.2 by considering AHR of 0.63,probability of event (neonate initiated TF) was 0.85 [29], two-sided,5% significant level, power of 80% and 10% probability of withdrawing from a study. After checking all significant factors from the previous study the maximum sample size obtained was 193. The total sample size was proportionally allocated to each study hospitals.

### Sampling procedure

The study area was chosen using a simple random sampling method based on a lottery approach from the 12 general hospitals: Aksum St. Mary, Suhul, Adigrat, Maychew, Miyani Sheraro, Adwa, Temben, Wukro,Mekelle, Quiha,

Alamata,Miyani, and Korem general hospitals. The hospitals included in the study are Aksum St. Mary Hospital, Shire Suhul Hospital, Adigrat Hospital, Maychew Hospital, Miyani Sheraro Hospital, and Adwa Hospital.

Over the past two months, the total number of preterm neonates admitted to the NICU ward in each selected general hospital was reviewed. The admissions recorded were as follows: Aksum St. Mary Hospital (66), Shire Suhul Hospital (70), Adigrat Hospital (64), Maychew Hospital (55), Miyani Sheraro Hospital (60), and Adwa Hospital (60).

From December 20, 2023, to February 30, 2024, a total of 375 preterm neonates were admitted to the NICU ward across the six selected public general hospitals. The samples were allocated proportionally to each hospital: 34 from Aksum St. Mary Hospital, 36 from Shire Suhul Hospital, 33 from Adigrat Hospital, 28 from Maychew Hospital, 31 from Miyani Sheraro Hospital, and 31 from Adwa Hospital. The final sample size was 193.

After allocation, finally, a systematic random sampling method ($K = N/n = 375/193 \approx 1.9 \approx 2$) was used. The first participant was selected randomly, and then every Kth interval (every 1st interval) was chosen as a study participant during admission.

## Study variable

**Dependent variable.** Time to initiate TF

**Independent variable. Maternal sociodemographic factors** such as age of the mother, residence, educational status and, maternal obstetric and medical related factors, mode of delivery, and maternal HIV AIDS status.

**Neonatal related factors**: birth weight, sex, gestational age, first -minute Apgar score, fifth- minute Apgar score, perinatal asphyxia, sepsis, hemodynamic instabilities, respiratory distress syndrome, weight for gestational age, and hypothermia.

**Health service- related factors**; place of delivery and KMC service.

## Operational definitions

**Trophic feeding**: The first minimal enteral feeding administering small volumes of enteral feeds (typically 10–20 mL/kg/day) via orogastric or nasogastric to stimulate the development of the immature gastrointestinal tract in preterm given to stimulate the gut [31].

**Early initiation of trophic feeding**: Preterm neonates start TF within 24 hours of birth [29].

**Delayed initiation of trophic feeding**: Preterm neonates start TF after 24 hours of birth [29].

**Time to initiate TF**: the length of time in hours, starting from birth to the first TF [10].

**Event:** preterm neonates who had initiated the first TF during the follow-up period [10].

**Censored**: Preterm neonates who died, left against medical advice, transferred or referred before starting TF, or not started at end follow-up [14].

**Follow up time**: time from birth up to either the study subjects' start TF or censored [10].

**Hemodynamic instabilities**: Blood group and RH incompatibility, anemia, polycythemia, bleeding disorders, and blood glucose disturbances [10].

## Data collection tools and procedure

Data collection was conducted using a semi-structured, pretested questionnaire specifically designed to ensure both reliability and relevance. This instrument encompassed a comprehensive range of topics, including health service utilization, key determinants of maternal and neonatal health, and core sociodemographic variables. The questionnaire's content was thoughtfully adapted and synthesized from a diverse body of established literature, ensuring it captured contextually significant and evidence-based insights relevant to the study objectives [10,14,29,31].

 

Through interviews, chart reviews with prospective follow-up, six nurses were involved in the data collection supervised by two supervisors. Data collectors (nurses working out of the institutions) and two supervisors were trained to follow up and itemize the included questionnaire, obtain client consent, and collect data accordingly. The supervisor reviewed the questionnaire for completeness.

## Data quality assurance

Before the actual data collection the questionnaire were adapted first in English, then translated to the local language Tigrigna, and then back to English to ensure consistency. Data collectors received two days training about the study and the content of the instrument in order to familiar about each question and follow up and it was also a mechanism of minimizing bias during the process of data collection.

During data collection, supervisors checked data for its completeness and missing information at each point. A pretest was done at Wukro K/awlaelo general hospital on 5% sample size.

## Data processing and analysis

Data were entered into Epi Data Manager V 4.6 after being checked for completeness and consistency. It was subsequently exported to STATA version 14.1 for cleaning, coding, and analysis. During this process, the level of missing values, the presence of multicollinearity were reviewed, and the value was 2.01, along with the Cox proportional hazards regression model assumptions were checked using the Schoenfeld residual test, which yielded a value of 0.12, greater than 0.05, indicating that the assumptions were fulfilled.

To estimate survival time and cumulative probabilities of initiating TF, descriptive statistics, Kaplan-Meier survival curves, life tables, and hazard functions were applied. Additionally, the log-rank test was used to compare the survival curves of different categories. Finally, bi-variable cox regression analysis were conducted and -value of < 0.25 were used as a cut-off point to enter variables to multivariable Cox-regression and, a test was consider significant at P. value < 0.05.

## Ethical clearance

An official ethical clearance letter was obtained from the Axum University College of Health Sciences Institutional Review Board (IRB) on 03 December 2024 with Protocol of Approval AKU-IRB 095/2024. Additionally, permission was secured from the Tigray Regional Health Bureau and the respective hospital medical director's office. Following these steps, written consent was waived from the parent/guardian of the preterm neonate, following a thorough explanation of the study's purpose, procedures, and potential risks and documentation of this consent has been recorded by the researcher for ethical compliance Upon receiving approval, communication was established with the medical directors, chief nursing directors, and intensive care unit heads at each general hospital.

Throughout the process, greatest care was taken to ensure patient confidentiality. Given the prospective follow up nature of the study harm to individual patients was minimal as long as confidentiality was strictly maintained. To guarantee confidentiality, all collected data were anonymized using codes and stored securely in a locked room before being entered into a password-protected computer system. Patient names were excluded from the data collection format entirely. The principal investigator remained the sole individual with access to the data. All methods were performed in accordance with declarations of Helsinki.

## Result

A total of 193 preterm neonates were included in the final analysis with a 100% response rate. Of these, 174 (87.6%) were started on TF, and 20 (10.4%) were censored. Most mothers were from urban areas (121, 62.7%), while 72 (37.3%) were

rural residents. Maternal age ranged from 18 to 38 years, with a mean of 25.72 years, and 77 (39.9%) had completed secondary education. Among the neonates, 120 (62.2%) had a gestational age ≥ 34 weeks. All were low birth weight, with a mean birth weight of 2300 g (Table 1).

**Table 1.** Socio demographic of mother and clinical characteristics of neonates admitted in NICU in public general hospitals of Tigray Ethiopia 2024/25.

| Socio-demographic, health care and clinical characteristics | Category | Status<br>Start TF censored | Total |
|---|---|---|---|
| Maternal age | 18-24 | 106(93.8%) 7(6.1%) | 113 |
| | 25-34 | 58(80.6%) 6(9.4%) | 64 |
| | >=35 | 9(76.2%) 7(43.8%) | 16 |
| Maternal education | Illiterate | 16(76.2%) 5(23.8%) | 21 |
| | Primary | 62(92.5%) 5(7.5%) | 67 |
| | Secondary | 72(93.5%) 5(6.5%) | 77 |
| | College and above | 23(82.1%) 5(17.9%) | 28 |
| Residence | Urban | 108 (89.2%) 13(10.8%) | 121 |
| | Rural | 65 (90.3%) 7(9.7%) | 72 |
| Birth weight in gram | 1500-2499 gm | 149(92.5%) 12(7.5%) | 161 |
| | <1500 gm | 24(75%) 8(25%) | 32 |
| Gestational age in weeks | >=34 | 111(92.5%) 9(7.5%) | 120 |
| | <33 | 62(84.9%) 11(15.1%) | 73 |
| 1st minute APGAR score | <7 | 96(86.4%) 15(13.5%) | 111 |
| | >=7 | 77(93.9%) 5(6.1%) | 83 |
| 5th minute APGAR score | <7 | 47(79.7%) 12(20.3%) | 59 |
| | >=7 | 126(94%) 8(6%) | 134 |
| Sex | Male | 51(91.1%) 5(8.9%) | 56 |
| | Female | 122(91%) 15(9%) | 137 |
| Perinatal Asphyxia | Yes | 69(88.5%) 9(11.5%) | 78 |
| | No | 104(90.4%) 11(9.6%) | 115 |
| Hemodynamic instability | Yes | 51(78.5%) 14(21.5%) | 65 |
| | No | 122(95.3%) 6(4.7%) | 128 |
| Respiratory Distress Syndrome | Yes | 50(76.9%) 15(23.1%) | 65 |
| | No | 123(96.8%) 5(3.2%) | 127 |
| Weight for Gestational Age (GA) | AFGA | 144(90.5%) 15(9.5%) | 159 |
| | SFGA | 29(85.3%) 5(14.7%) | 34 |
| Mode delivery | SVD | 154(92.2%) 13(7.8%) | 167 |
| | CS | 19(73.1%) 7(26.9%) | 26 |
| Place of delivery | In born | 115(93.5%) 8(6.5%) | 123 |
| | Out born | 64(90.1%) 7(9.9%) | 71 |
| Sepsis | Yes | 145(90.6%) 15(9.4%) | 160 |
| | No | 28(84.8%) 5(15.2%) | 33 |
| Hypothermia | Yes | 132(90.4%) 14(9.6%) | 146 |
| | No | 41(88.2%) 6(12.8%) | 47 |
| | | | |
| Kangaroo mother care (KMC) | Yes | 98(93.3%) 7(6.7%) | 105 |
| | No | 75(85.2%) 13(14.8%) | 88 |

## Survival status of neonates on time to initiate TF

A total of 8,382 person-hours of risk time were followed for 193 pairs of newborns and mothers. The follow-up period ranged from 12 hours to 84 hours. Out of 193 preterm newborns, 173 (8 95% CI: 89.67–89.73) were starting TF, and the remaining 20 (10.4%) were censored (Fig 1).

The median time to initiate TF for the entire follow-up study was 45 hours (95% CI: 42–56) (interquartile range: 24–65), and the median follow-up time was 43 hours (interquartile range: 24–60). In the first day (24 hours), only 58 (30%) preterm neonates initiated TF, and the remained 116 (58.5%) and 166 (86%) started by the end of 48 and 72 hours. Finally, only 8 neonates remained, with 7 starting TF and 1 being censored.

The cumulative survival probabilities of preterm neonates by the end of 24, 48, 72, and 84 hours were 69.7% (95% CI: 0.6267–0.7569), 39.6% (95% CI: 0.3256–0.4648), 8% (95% CI: 0.0458–0.1273), and 0.5% (95% CI: 0.0002–0.0450), respectively (Table 2).

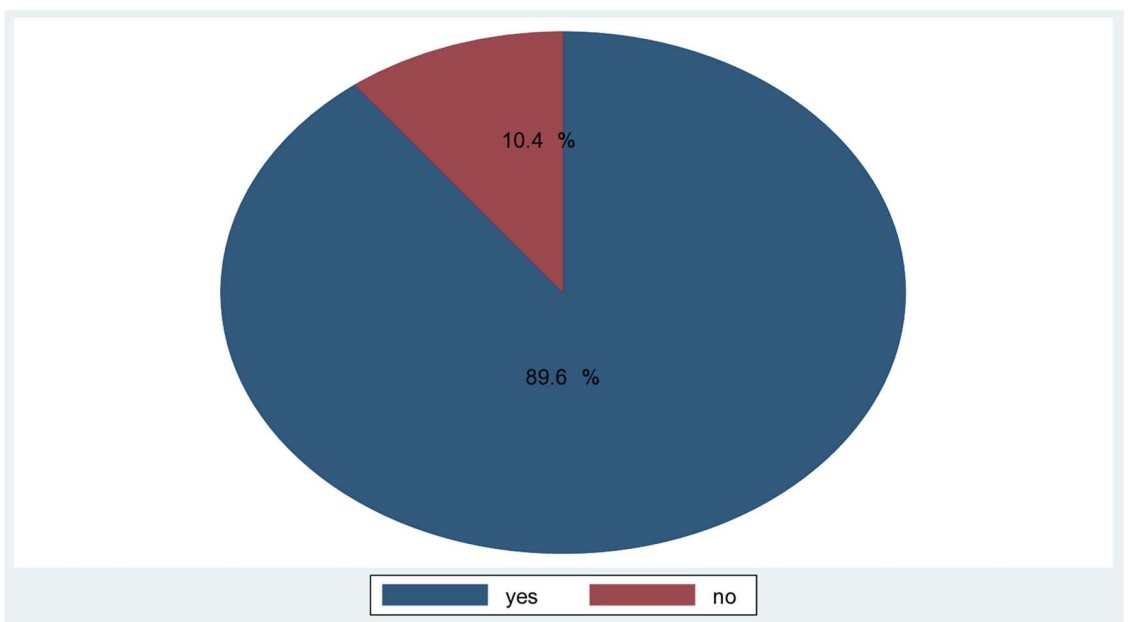

**Fig 1. Proportion of initiated trophic feeding and censored among preterm neonates admitted in NICU at General public Hospitals in Tigray, Ethiopia, 2024/25.**

**Table 2. Life table analysis of time to initiate trophic feeding among preterm neonates admitted in NICU in General public Hospitals from December 20, 2024 to February 30, 2025.Tigray, Ethiopia.**

| Interval | Beginning Total | Start TF | Censored | Survival Probability | 95% Confidence Interval |
|---|---|---|---|---|---|
| 0-24 | 193 | 58 | 5 | 0.6971 | 0.6267 - 0.7569 |
| 24-48 | 130 | 55 | 3 | 0.3957 | 0.3256 - 0.4648 |
| 48-72 | 72 | 53 | 11 | 0.0803 | 0.0458 - 0.1273 |
| 72-96 | 8 | 7 | 1 | 0.0054 | 0.0002 - 0.0450 |

## Comparison of survival among different categorical variables

Kaplan–Meier curves were constructed to compare the overall survival patterns over time between different groups. Similarly, the log-rank test analysis indicated that there were substantial differences in median time to initiate TF among preterm neonates across categories of several variables (Fig 2).

Accordingly, there was a significant difference in median time to initiate TF: for BW 1500–2499 grams was 36 hours (95% CI: 33–39) compared to BW less than 1500 grams, 72 hours (95% CI: 71–73), which was statistically significant (log-rank test = 60, P < 0.001). Neonates with GA ≥ 34 weeks exhibited significantly earlier onset at 27 hours (95% CI: 25–29) compared to 60 hours (95% CI: 53–67) for those with GA < 34 weeks. Concerning clinical-related factors of preterm neonates, those who had RDS experienced a significantly delayed TF initiation of 70 hours (95% CI: 68–72) compared to those without RDS, 27 hours (95% CI: 24–30), and this was statistically significant (log-rank test = 86.6, P < 0.001).

Similarly, the median time to initiate TF was significantly different for neonates with a first-minute APGAR score < 7, at 60 hours (95% CI: 57–63), compared to a first-minute APGAR score ≥ 7, at 24 hours (95% CI: 23.8–24.2), at P < 0.001. In a similar vein, neonates with a fifth-minute APGAR score < 7 minutes had 72 hours of median time to start TF (95% CI: 71–73) compared to a fifth-minute APGAR score ≥ 7 minutes, at 30 hours (95% CI: 25–35), with P < 0.001. Finally, neonates without KMC had late commencement of TF, with a 65-hour median time to start TF (95% CI: 61–69) compared to neonates with KMC, 24 hours (95% CI: 23.7–24.3), which was statistically significant at P < 0.001 (Table 3).

## Predictors of time to initiate trophic feeding in preterm neonates

Variables included in the bi-variate analysis with P value < 0.25 were BW, GA, first-minute APGAR score, 5th-minute APGAR score, hemodynamic instability, RDS, weight for gestational age, sepsis, hypothermia, asphyxia, KMC, mode of delivery, and place of delivery.

Hence, the multivariate analysis revealed that the hazard of initiating TF among preterm neonates with a birth weight of less than 1500 grams was 84% lower compared to those born with a weight between 1500–2499 grams (AHR: 0.16, 95% CI: 0.075–0.355). Preterm neonates who had a first-minute APGAR score below seven were 54% less likely to start TF compared to those with an APGAR score of seven or higher (AHR: 0.46, 95% CI: 0.273–0.784). Similarly, the likelihood of

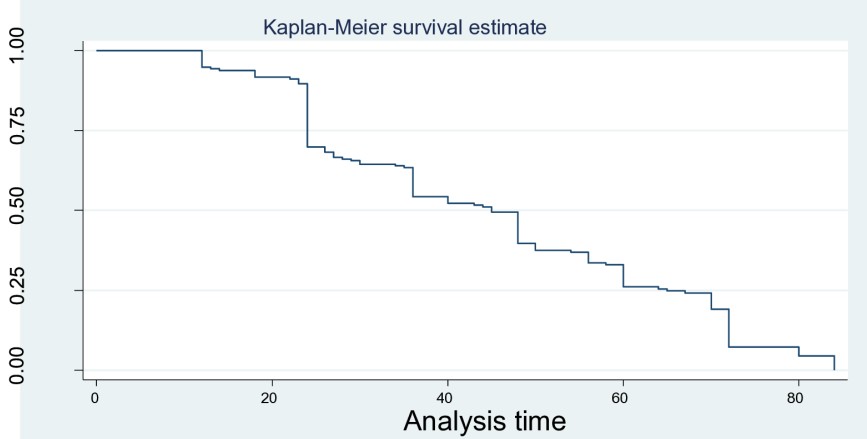

**Fig 2. Over all Kaplan Meier estimation time to initiate trophic feeding and its predictors among preterm neonates admitted in NICU in general public hospitals from December 20, 2024 to February 30, 2025, Tigray, Ethiopia.**

**Table 3. Survival time and log rank analysis of preterm neonates according to different clinical characteristics of neonates admitted to public General hospitals in NICU from December 20, 2024 to February 30, 2025, in Tigray, Ethiopia.**

| Covariates | Category | Median Survival Time In Hours (95% Ci) | Log Rank (X2 Value) | P-Value |
|---|---|---|---|---|
| Birth Weight In Gram | 1500-2499 | 36(33-39) | 60 | <0.001 |
| | <1500 | 72(71-73) | | |
| Gestational Age In Weeks | >=34 | 27(25-30) | 51.9 | <0.001 |
| | <34 | 60(53-67) | | |
| 1st Minute Apgar Score | <7 | 60(57-63) | 140 | <0.001 |
| | >7 | 24(23.8-24.2) | | |
| 5th Minute Apgar Score | <7 | 72(71-73) | 115 | <0.001 |
| | >=7 | 30(25-35) | | |
| Perinatal Asphyxia | Yes | 60(54-66) | 42.2 | <0.001 |
| | No | 27(24-30) | | |
| Hemodynamic Instability | Yes | 70(64-76) | 51.6 | <0.001 |
| | No | 30(25-35) | | |
| Weight For Gestational Age | AFGA | 36(33-39) | 33.8 | <0.001 |
| | SFGA | 72(72-73) | | |
| RDS | Yes | 70(68-72) | 87.2 | <0.001 |
| | No | 27(24-30) | | |
| Place Of Delivery | In Born | 27(25-29) | 95.6 | <0.001 |
| | Out Born | 70(65-75) | | |
| Sepsis | Yes | 48(45-51) | 20.8 | <0.001 |
| | No | 24(23.8-24.2) | | |
| Hypothermia | Yes | 48(41.1-58.9) | 11.5 | <0.001 |
| | No | 24(22.7-25.2) | | |
| Kangaroo Mother Care (KMC) | Yes | 24(23.7-24.3) | 132 | <0.001 |
| | No | 65(61-69) | | |

initiating TF in preterm neonates with a fifth-minute APGAR score below seven was 62% lower compared to those with a score of seven or higher (AHR: 0.38, 95% CI: 0.211–0.679).

Additionally, preterm neonates with respiratory distress syndrome (RDS) were 59% less likely to start TF compared to those without RDS (AHR: 0.41, 95% CI: 0.25–0.66). Finally, the hazard of initiating TF in the neonate without KMC was linked to a 59% lower chance of initiating TF earlier compared to those who received KMC (AHR: 0.41, 95% CI: 0.217–0.77) (Table 4).

The Cox proportional hazards assumption was evaluated using the overall global test for the full model and showed no violation (P = 0.12). All covariates met the proportional hazards assumption, and goodness of fit was assessed using Cox–Snell residuals (Fig 3).

## Discussion

This study aimed to assess the time to initiate TF and its predictors among preterm neonates admitted to the study hospital within the study period. In this study, the incidence of starting TF was 2 per 100 person-hours of risk time. At the end of follow-up, 173 (89.6%) of neonates (95% CI: 84.4–93.4) were started on TF, and 20 (10.4%) were censored.

This finding on the proportion of admitted preterm neonates who initiated TF is consistent with the study finding in, Addis Ababa 85%, [29]. This may be due to the similarity in study design, which is a prospective cohort, and sample

**Table 4. Results of bivariate and multivariate Cox regression analysis of time to initiate TF and its predictor factor among preterm neonates admitted at public general Hospitals in NICU from December 20,2024 to February 30,2025,Tigray,Ethipia (n = 193).**

| Covariates | Category | Started TF Yes No | Crude Hazard Ratio (CHR) | P-Value | Adjusted Hazard Ratio (AHR:95% Ci) | P-Value |
|---|---|---|---|---|---|---|
| Hemodynamic Instability | Yes | 51 14 | 0.34(0.24-0.48) | <0.001 | 0.78(0.49-1.22) | 0.26 |
| | No | 122 6 | 1 | | 1 | |
| Perinatal Asphyxia | Yes | 69 9 | 0.39(0.28-0.54) | <0.001 | 0.66(0.42-1.03) | 0.068 |
| | No | 104 11 | 1 | | 1 | |
| Sepsis | Yes | 145 15 | 1 | | | |
| | No | 28 5 | 2.4(1.6-3.7) | <0.001 | 0.77(0.42-1.230 | 0.31 |
| Hypothermia | Yes | 132 14 | 0.57(0.4-0.8) | 0.002 | 0.79(0.5-1.2) | 0.30 |
| | No | 41 6 | 1 | | 1 | |
| Weight For Gestational Age | AFGA | 144 15 | 1 | | | |
| | SFGA | 29 5 | 0.3(0.2-0.5) | <0.001 | 1.5(0.8-3.07) | 0.17 |
| Mode Of Delivery | Vaginal | 154 13 | 1.4(0.91-2.39) | 0.1 | 1.6(0.94-2.95) | 0.078 |
| | Cesarean Section | 19 7 | 1 | | | |
| Place Of Delivery | In Born | 114 8 | 1 | | | |
| | Out Born | 59 12 | 0.2(0.14-0.30) | <0.001 | 0.66(0.35-1.2) | 0.19 |
| Gestational Age In Weeks | >=34 | 111 9 | 1 | | | |
| | <34 | 62 11 | 0.32 (0.22-0.43) | <0.001 | 0.69(0.44-1.09) | 0.11 |
| Birth Weight In Gram | 1500-2499 | 149 12 | 1 | | 1 | |
| | <1500 | 24 8 | 0.19 (0.12-0.32) | <0.001 | 0.16(0.08-0.355) | **<0.001** |
| First-Minute Apgar Score | <7 | 96 15 | 0.16 (0.11-0.22) | <0.001 | 0.46(0.27-0.78) | **0.004** |
| | >=7 | 77 5 | 1 | | 1 | |
| Fifth-Minute Apgar Score | < 7 | 47 12 | 0.14 (0.09-0.2) | <0.001 | 0.38(0.21-0.68) | **0.001** |
| | >= 7 | 126 8 | 1 | | | |
| Respiratory Distress Syndrome (RDS) | Yes | 50 15 | 0.23(0.16-0.33) | <0.001 | 0.41(0.25-0.66) | **<0.001** |
| | No | 123 5 | 1 | | 1 | |
| Kangaroo Mother Care (KMC) | Yes | 98 7 | 1 | | | |
| | No | 75 13 | 0.15(0.11-0.22) | <0.001 | 0.41(0.22-0.77) | **0.006** |

size (193 vs 153). However, it is higher than the studies done in southern Oromia (73.2%) [10]. This discrepancy may be attributed to differences in the number of study areas and research design; prospective cohort studies facilitate real-time data collection, whereas retrospective cohort studies depend on pre-existing records and the number of study areas [32].

In this study 58 (30%) of preterm neonates started TF within the first 24 hours of birth, 58.5% and 86% of preterm neonates started TF within 48 and 72 hours of birth, respectively. This indicates that only a small proportion of preterm neonates started TF within the first 24 hours of birth. However, this finding is lower than those reported in studies conducted in Nigeria and Kenya 48% and retrospective study in Portugal 44% observational study in Italy 74.1% [13,33,34].

The disparity in the initiation of TF among preterm neonates can be attributed to several factors, including healthcare infrastructure, provider knowledge on training and understanding of feeding protocols, and access to care, and socioeconomic conditions with better specialized facilities and resources, tend to show higher rates of early feeding [35].

In this prospective study, the median surviving time to initiate TF was 45 hours (95% CI: 42–56). This is similar study done in southern Oromia 48 hours [10]. but, higher than the finding reported in a prospective follow up study done in

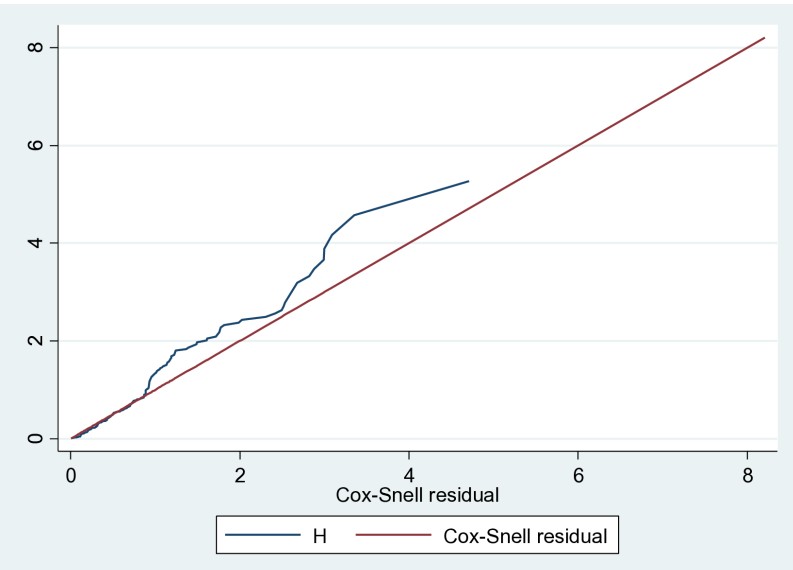

**Fig 3. Nelson Aalen cumulative hazard graph against cox Snell residual on time to initiate trophic feeding and its predictors among preterm neonate admitted in public general hospitals from December 20, 2024 to February 30,2025,Tigray Ethiopia**

Addis Ababa 41 hours [29] and retrospective study done in North West Ethiopia 42 hours [14]. The discrepancy might be regional healthcare practices, availability of specialized facilities, local policies regarding neonatal care, variability on providing TF feeding during the neonatal period and differences in healthcare infrastructure across regions.

In addition, this result is lower than retrospective studies conducted in Portugal 72 hours [33].The difference might be due to sample size (193 vs 219), study period, study population, population difference, and variation in neonatal management protocols (Ethiopia VS Portugal).

In this study, preterm neonates with a very low birth weight < 1500 g were 84% less likely to initiate TF on time compared to those with a BW > 1500 g. This finding is higher than with the results of a retrospective follow-up study conducted in southern Oromia, Ethiopia, 55% [10]. This could be The conflict in Tigray has severely damaged health infrastructure, this destruction, combined with increased poverty and economic instability, Poor maternal nutrition and lack of healthcare access further exacerbate this issue, resulting in worse health outcomes compared to regions like Oromia, where healthcare resources are more available and significantly impacts the health of vulnerable newborns in Tigray [36,37]. Similarly, this could be underdevelopment of organs and lower readiness for enteral feeding in preterm neonates with lower birth weight [2].

Preterm neonates with an APGAR score below seven at one minute had a 54% lower likelihood of starting TF compared to those with a score of seven or higher. This finding is consistent with a retrospective study conducted in Northwest Ethiopia 40% [14] and prospective study in Addis Ababa 60% [29].

Similarly, Preterm neonates with a fifth-minute APGAR score below seven were 62% less likely to begin TF compared to those with an APGAR score of seven or higher. This also supported by study conducted in Addis ababa,Ethipia 49% [29]. This similarity could be due to a low APGAR score, which indicates potential intrauterine hypoxia and this condition suggests reduced blood flow to the gastrointestinal tract, potentially leading to severe NEC and resulting in feeding intolerance [38].

The hazard of initiating TF in preterm neonates with RDS had 59% less likely of initiating TF compared to those without the condition. This result aligns with a study conducted in the North West, which reported a 50% reduction [14] and Addis

Ababa, Ethiopia (37%) [29]. A possible explanation for this might be neonates with RDS often have rapid breathing, which may be due worsened by pressure in the abdominal, further compounded by physiological instability which heightens the risk of aspiration, feeding intolerance, and vomiting [39]. While this may lead professionals to delay starting tube feeding, global guidelines recommend that minimal tube feeding can be initiated within the first 24 hours of life for neonates with RDS [40].

Finally, WHO advises that KMC should be given to preterm or low birth weight infants, starting either in healthcare settings or at home, for a duration of 8–24 hours daily [41]. Based on this evidence, our study revealed that not having KMC was associated with a 59% reduced chance of starting TF early compared to those who did receive KMC. This finding is similar with a retrospective cohort study done in Oromia, Ethiopia 42% [10] and a retrospective follow-up study in Turkey identified KMC as a key factor in enhancing enteral feeding skills among preterm neonates [42]. Additionally, a prospective cohort study conducted in a teaching hospital in India found that early implementation of KMC was safe and correlated with a shorter time to reach full feeds in preterm infants [43]. Furthermore, a meta-analysis of randomized controlled trials indicated that KMC promotes the early initiation of breastfeeding among preterm and low birth weight infants [44].This might be KMC improved survival rates, reduced infections, enhanced feeding and growth for the baby as well as bonding and maternal confidence [45].

Generally, the study shows that TF often started much later than the recommended 24 h, pointing to the need for NICUs to review and strengthen their early feeding routines. Despite the critical need for structured feeding strategies, many hospitals currently lack established protocols for trophic feeding. As a result, the initiation and management of trophic feeding in preterm neonates often vary widely among healthcare providers, leading to inconsistent practices. This study highlights several critical implications for clinical practice, public health policy, and future research related to feeding initiation in preterm neonates. The significant challenges observed in initiating TF, particularly in very low birth weight infants and those with low APGAR scores and RDS necessitate the development of tailored feeding protocols and prompt early interventions. Additionally, Public health policies should focus on comprehensive maternal education and nutrition to improve outcomes, while training programs for healthcare providers should emphasize the implementation of KMC to promote early feeding and bonding.

### Limitations of the study

The study requires continuous monitoring of neonates, repeated clinical assessments, dedicated staff involvement, and systematic data management throughout the follow-up period and these requirements may increase financial costs and place additional workload on NICU staff, potentially affecting data collection period.

### Conclusion

The study revealed a lag in starting TF for preterm neonates compared to standard national guidelines, underscoring the need to improve early feeding practices for this vulnerable population. Delays in initiating TF can negatively affect neonatal health outcomes, potentially leading to a higher risk of infections or longer hospital stays. Only 30% of infants received TF within the first 24 hours, indicating a gap between current practices and the recommended protocols. Once potential confounding factors were taken into account, key predictors of delayed TF initiation included very low birth weight, APGAR scores below seven at one and five minutes, the presence of RDS, and the absence of KMC.

Preventive measures can significantly enhance the likelihood of timely TF initiation for vulnerable infants. Key strategies include educating caregivers about optimal feeding practices, promoting immediate skin-to-skin contact through KMC to encourage bonding and stimulate feeding instincts, and implementing proactive monitoring to assess infant health and readiness for feeding. Additionally, establishing multidisciplinary care teams that involve pediatricians, nurses, and lactation consultants ensures comprehensive support, while tailored nutritional support protocols for very low birth weight and low APGAR score infants can facilitate a smoother transition to feeding. By focusing on these preventive measures,

healthcare providers can create an environment that reduces risks associated with delayed feeding and improves overall health outcomes for infants.

## Recommendations

**Enhanced Monitoring of VLBW Infants:** Implement protocols for early and continuous monitoring of VLBW preterm infants to address feeding challenges proactively.

**Immediate Postnatal Care Interventions:** Focus on improving APGAR scores through immediate resuscitation and supportive care in the delivery room to ensure infants start on a stable footing.

**Management of RDS:** Develop integrated care pathways that streamline respiratory support and feeding interventions for infants diagnosed with RDS to facilitate earlier feeding initiation.

**Promote Kangaroo Mother Care (KMC):** Encourage and train caregivers on the importance of KMC immediately post-birth. Initiating skin-to-skin contact can improve early feeding success and overall infant health outcomes.

**Ministry of Health**: Enhance and develop early feeding protocols and implement standardized protocols to promote the initiation of enteral feeding within the first 24 hours of birth, especially for neonates with stable vital signs.

Finally, researchers should conduct long-term studies to assess the impact of early TF initiation on growth and neurodevelopmental outcomes in preterm infants in this area and to identify barriers to timely initiation of trophic feeding in various healthcare settings to guide to best practices for early TF initiation in preterm neonates and evaluate the impact of guideline adherence on short- and long-term outcomes in preterm neonates and explore effective interventions that can promote early initiation of trophic feeding in line with national practices.

## Supporting information

**S1 File. Dataset.**
(XLSX)

## Acknowledgments

I would like to express my sincere gratitude to my data collectors and supervisors for their facilitation, organization and collection of the data throughout the study period.

## Author contributions

**Conceptualization:** Teklebrhan Welderufael Kidane, Zeray Baraki, Asefa Iyasu.

**Data curation:** Teklebrhan Welderufael Kidane, Tekle Gebremeskel Ygzaw, Binyam Gebrehiwet Tesfay, Ngsti Gebremichael Beyene, Geberziher Gebreslassie Welearegay.

**Formal analysis:** Teklebrhan Welderufael Kidane, Zeray Baraki, Asefa Iyasu, Tekle Gebremeskel Ygzaw, Binyam Gebrehiwet Tesfay, Ngsti Gebremichael Beyene, Geberziher Gebreslassie Welearegay.

**Investigation:** Teklebrhan Welderufael Kidane, Zeray Baraki, Asefa Iyasu.

**Methodology:** Teklebrhan Welderufael Kidane.

**Project administration:** Teklebrhan Welderufael Kidane.

**Software:** Teklebrhan Welderufael Kidane.

**Supervision:** Teklebrhan Welderufael Kidane, Zeray Baraki, Asefa Iyasu.

**Validation:** Teklebrhan Welderufael Kidane.

**Writing – original draft:** Teklebrhan Welderufael Kidane, Binyam Gebrehiwet Tesfay, Teklewoini Mariye Zemicheal, Nebiat Desale Gidey.

**Writing – review & editing:** Teklebrhan Welderufael Kidane, Zeray Baraki, Asefa Iyasu, Binyam Gebrehiwet Tesfay, Teklewoini Mariye Zemicheal, Nebiat Desale Gidey.

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
