## [Decision Letter · Decision Letter 0]

12 Jan 2026

Dear Dr. kidane,

Thank you for submitting your manuscript to PLOS ONE. After careful consideration, we feel that it has merit but does not fully meet PLOS ONE’s publication criteria as it currently stands. Therefore, we invite you to submit a revised version of the manuscript that addresses the points raised during the review process.

Please submit your revised manuscript by Feb 26 2026 11:59PM. If you will need more time than this to complete your revisions, please reply to this message or contact the journal office at plosone@plos.org . A letter that responds to each point raised by the academic editor and reviewer(s). You should upload this letter as a separate file labeled 'Response to Reviewers'.A marked-up copy of your manuscript that highlights changes made to the original version. You should upload this as a separate file labeled 'Revised Manuscript with Track Changes'.An unmarked version of your revised paper without tracked changes. You should upload this as a separate file labeled 'Manuscript'.

We look forward to receiving your revised manuscript.

Kind regards,

Yusuke Hoshino

Academic Editor

PLOS One

Journal Requirements:

[The authors have declared that no competing interests exist.].

We note that one or more of the authors are employed by a commercial company: Araya Kahsu College of Health Science and Medicine.

5. Please ensure that you refer to Figures 1, 2, 3, and 4 in your text as, if accepted, production will need this reference to link the reader to the figures.

6. Please include a separate caption for each figure in your manuscript.

7. We note you have included a table to which you do not refer in the text of your manuscript. Please ensure that you refer to Tables 1, 2, 3, and 4 in your text; if accepted, production will need this reference to link the reader to the tables.

8. We note that there is identifying data in the Supporting Information file <Dataset.xlsx>. Due to the inclusion of these potentially identifying data, we have removed this file from your file inventory. Prior to sharing human research participant data, authors should consult with an ethics committee to ensure data are shared in accordance with participant consent and all applicable local laws.

-Location data

Please remove or anonymize all personal information (Age), ensure that the data shared are in accordance with participant consent, and re-upload a fully anonymized data set. Please note that spreadsheet columns with personal information must be removed and not hidden as all hidden columns will appear in the published file.

Additional Editor Comments:

Please provide a point-by-point response to each of the reviewers’ comments, indicating clearly how you have addressed each concern in your revised manuscript.

Reviewers' comments:

Reviewer's Responses to Questions

**Comments to the Author**

1. Is the manuscript technically sound, and do the data support the conclusions?

Reviewer #1: Yes

Reviewer #2: Yes

2. Has the statistical analysis been performed appropriately and rigorously?

Reviewer #1: Yes

Reviewer #2: Yes

3. Have the authors made all data underlying the findings in their manuscript fully available?

Reviewer #1: Yes

Reviewer #2: No

4. Is the manuscript presented in an intelligible fashion and written in standard English?

Reviewer #1: Yes

Reviewer #2: No

Reviewer #1: This is an automated report for PONE-D-25-54861. This report was solicited by the PLOS One editorial team and provided by ScreenIT.

ScreenIT is an independent group of scientists developing automated tools that analyze academic papers. A set of automated tools screened your submitted manuscript and provided the report below. Each tool was created by your academic colleagues with the goal of helping authors. The tools look for factors that are important for transparency, rigor and reproducibility, and we hope that the report might help you to improve reporting in your manuscript. Within the report you will find links to more information about the items that the tools check. These links include helpful papers, websites, or videos that explain why the item is important. While our screening tools aim to improve and maintain quality standards they may, on occasion, miss nuances specific to your study type or flag something incorrectly. Each tool has limitations that are described on the ScreenIT website. The tools screen the main file for the paper; they are not able to screen supplements stored in separate files. Please note that the Academic Editor had access to these comments while making a decision on your manuscript. The Academic Editor may ask that issues flagged in this report be addressed. If you would like to provide feedback on the ScreenIT tool, please email the team at ScreenIt@bih-charite.de. If you have questions or concerns about the review process, please contact the PLOS One office at plosone@plos.org.

Reviewer #2: Manuscript title: "Trophic Feeding Initiation and its Predictors in Preterm Neonates Admitted to Neonatal Intensive Care Units

Preterm enteral nutrition feeding is an important topic with significant variability despite multiple recommendations. I applaud the authors’ efforts to analyze trophic feeding initiation.

Abstract:

Introduction: in the abstract should be a short convictive statement showing the identified gap.

Conclusion: which factors are risk and which are preventive should be concluded. Make the conclusion sound with specific recommendations.

AIM and primary outcomes not defined

Ethics: is written informed consent waved or is it signed? it should clearly be stated. If signed who signed the consent?

Introduction:

The introduction appears vague and extensive. It should focus on what is known of the problem and what gaps this study looks to evaluate and solve.

Line 62-67-paraphrase the idea

Please describe, in a succinct way, the presence of feeding protocols at the hospitals

Please summarize line 120-123 including reference citations

The manuscript from line 129 -134 is not convective to conduct this study. Please revise it. What gaps were identified and what was your study’s main aim?

What other studies found regarding your topic and what is left should be explained.

Generally, introduction section should focus more on reasoning’s why providers do not start trophic feeds early despite the evidence available. Also, state how this manuscript will contribute to the evidence available regarding this topic.

Method:

Please define pre-diagnosed Stage II or III NEC, Stage III asphyxia and why is this an exclusion criterion

The Methods and materials section does not provide adequate information to replicate the study.

Simple definitions are not included, like: how are TF provided?

Line 227: Survival time is a confusing name for the time to start TF

Line 234-246 the idea on this text should be clear and summarized: and also please specify/clarify the purpose of the questionnaire; to whom it was administered? This goes accordingly with my comment regarding lack of clarity about the execution of the study.

Please omit line 274-278

Result:

Line 295-310 un-necessary redundant information since the information is provided with Table 1

Line 350-351 “Kaplan-Meier curves constructed to compare the overall survival patterns over time ‘’for’’ between different groups”. Please omit ‘’for’’

Discussion:

The discussion is focused on comparing the findings with those noticed in other studies. However, are the findings aligned with international literature?

Overall the discussion section needs major revision; does not clearly synthesize your findings and the implication of these findings.

The limitation section should be paraphrased

Regarding language: there are English grammatical and syntaxis errors in the whole document. Please ensure that, before resubmission, a person proficient in written English edits the manuscript.

**Do you want your identity to be public for this peer review?** For information about this choice, including consent withdrawal, please see our Privacy Policy

Reviewer #1: No

Reviewer #2: **Yes:** Mekonen Adimasu Kebede

---

## [Author Response · Author response to Decision Letter 1]

7 Feb 2026

Academic Editor comments

1.In the abstract should be a short convictive statement showing the identified gap

Answer:We accept the comment and we made corrections accordingly.

2.Please ensure that your manuscript meets PLOS ONE's style requirements, including those for file naming. The PLOS ONE style templates can be found at https://journals.plos.org/plosone/s/file?id=wjVg/PLOSOne_formatting_sample_main_body.pdf and https://journals.plos.org/plosone/s/file?id=ba62/PLOSOne_formatting_sample_title_authors_affiliations.pdf

Answer:We accept the comment and we made corrections according the PLOS ONE guideline

3.Your ethical statement should appear at the method section

Answer:We accept the and we brings it to the method section

4.Please include a separate caption for each figure

Answer:Thank you for the comment. We have included caption for each separate figures

5.We note that you have included a table to which you do not refer in the text of your manuscript

Answer:We accept the comment .we have shifted the text toward the table that represented

(Table 4. Results of bivariate and multivariate Cox regression analysis of time to initiate TF and its predictor factor among preterm neonates admitted at public general Hospitals in NICU from December 20,2024 to February 30,2025,Tigray,Ethipia.(n=193) towards the text represents.

6.Which factors are risk and which are preventive should be concluded. Conclude sound with specific recommendations

Answer:Thank you for the comment. Risk factors are : BW < 1500 gm, APGAR scores < 7 (both at 1 and 5 minutes), RDS

Preventable factors are: Key strategies promoting immediate skin-to-skin contact through Kangaroo Mother Care (KMC) and nutritional support protocols for very low birth weight and low APGAR score infants can facilitate a smoother transition to feeding

7.Is written informed consent waved or is it signed? It should clearly be stated. If signed who signed the consent?

Answer:Written consent was waived from the parent/guardian of the preterm neonate

8.Line 62-67-paraphrase the idea

Answer:The comment is well taken and we paraphrased the idea

9.Please describe, in a succinct way, the presence of feeding protocols at the hospitals?

Answer:We accept the comment. Despite the critical need for structured feeding strategies, many hospitals currently lack established protocols for trophic feeding. As a result, the initiation and management of trophic feeding in preterm neonates often vary widely among healthcare providers, leading to inconsistent practices.

10.Please summarize line 120-123 including reference citations?

Answer:We accept the comments and we made summarization on the idea

11.The manuscript from line 129 -134 is not convective to conduct this study. Please revise it. What gaps were identified and what was your study’s main aim?

Answer:The manuscript from line 129 -134 is not convective to conduct this study. Please revise it. What gaps were identified and what was your study’s main aim?

The comment is well taken. The timing of initiating TF in preterm neonates is critical for their health, yet significant gaps exist in understanding this process, especially in general hospitals in Tigray, Ethiopia. Current literature demonstrates limited evidence on the timely initiation of TF among preterm neonates admitted to Neonatal Intensive Care Units (NICUs), and many studies do not utilize a prospective follow-up methodology. This lack of localized data contributes to delays in TF initiation, potentially impacting health outcomes for these vulnerable infants. Identifying specific barriers that affect the timing of TF, such, resource constraints, and the training needs of healthcare providers, is essential for informing improvements in clinical practices.

12.What other studies found regarding your topic and what is left should be explained? Why providers do not start trophic feeds early despite the evidence available. Also, state how this manuscript will contribute to the evidence available regarding this topic?

Answer:recognizing the benefits of early TF, healthcare providers may still hesitate to start feeding promptly due to concerns about the stability of the infant, inadequate training in nutritional guidelines, and systemic barriers within the NICU setting. This study aims to investigate the predictors of timely TF initiation among preterm neonates in Tigray and will contribute valuable insights to the existing evidence base. By elucidating the factors that delay TF, the research can provide targeted recommendations to enhance care quality, reduce healthcare costs, and improve long-term health outcomes for families, thereby ensuring that preterm infants receive the critical nutrition they need on time.

13.Please define pre-diagnosed Stage II or III NEC, Stage III asphyxia and why is this an exclusion criterion?

Answer:Pre-diagnosed Stage II or III NEC and Stage III asphyxia are critical exclusion criteria in this study. NEC is a severe gastrointestinal condition predominantly affecting preterm infants, with Stage II indicating moderate disease characterized by abdominal distension and systemic illness, while Stage III represents a more severe form that may involve bowel perforation and increased risk of complications. Stage III asphyxia, or severe hypoxic-ischemic encephalopathy, results from significant oxygen deprivation at birth, leading to severe neurological impairment and clinical instability. Introducing trophic feeding in infants with these conditions could exacerbate their fragile physiological state and increase the risk of serious complications. By excluding these infants, the study can focus on a more homogeneous population of preterm neonates who are better equipped to tolerate early trophic feeding, thus yielding clearer insights into the predictors and benefits of timely TF initiation

14.The Methods and materials section does not provide adequate information to replicate the study?

Answer: Thank you for the comment. We incorporated the suggestions in the method section

15.Simple definitions are not included, like: how are TF provided?

Answer:The feeding can be administered via an orogastric or nasogastric tube, depending on the infant's condition and the clinical setting. This method ensures that the feeding is provided directly into the stomach

16.Line 227: Survival time is a confusing name for the time to start TF

Answer:The comment is well taken and I have replaced with Time to Initiate Trophic Feeding

17.Line 234-246 the idea on this text should be clear and summarized: and also please specify/clarify the purpose of the questionnaire; to whom it was administered? This goes accordingly with my comment regarding lack of clarity about the execution of the study?

Answer:Primarily used to gather information from respondents and provide standardized method to ask questions, ensuring consistency in response across different participants.

The questions was administered to their parents or guardians

18.Please omit line 274-278

Answer:Thank you for the nice comment and we omitted it.

19.Line 295-310 un-necessary redundant information since the information is provided with Table 1

Answer:Thank you for the nice comment and we omitted it.

20.Line 350-351 “Kaplan-Meier curves constructed to compare the overall survival patterns over time ‘’for’’ between different groups”. Please omit ‘’for’’

Answer:Thank you for the nice comment and we omitted ‘’for’’

21.The discussion is focused on comparing the findings with those noticed in other studies. However, are the findings aligned with international literature?

Answer:Thank for the comment.

Yes, we found findings is aligned with international studies. Example, a retrospective follow-up study in Turkey identified KMC as a key factor in enhancing enteral feeding skills among preterm neonates. Additionally, a prospective cohort study conducted in a teaching hospital in India found that early implementation of KMC was safe and correlated with a shorter time to reach full feeds in preterm infants. Furthermore, a meta-analysis of randomized controlled trials indicated that KMC promotes the early initiation of breastfeeding among preterm and low birth weight infants and we included in the manuscript discussion part.

22.Overall the discussion section needs major revision; does not clearly synthesize your findings and the implication of these findings

Answer:We accepted your comments and we made corrections accordingly

23.The limitation section should be paraphrased

Answer:We accept the comments and we paraphrased to Limitation of this study is the need for continuous monitoring and follow-up. This demand strains resources and necessitates additional time commitments from healthcare providers

24.Regarding language: there are English grammatical and syntaxis errors in the whole document. Please ensure that, before resubmission, a person proficient in written English edits the manuscript

Answer:We accepted your comment and we made corrections accordingly across all the manuscript.

---

## [Editor Report · Decision Letter 1]

3 Mar 2026

Time to initiate trophic feeding and predictors among preterm neonates admitted at General Hospitals in Tigray, 2025.

PONE-D-25-54861R1

Dear Dr. Teklebrhan welderufael kidane,

We’re pleased to inform you that your manuscript has been judged scientifically suitable for publication and will be formally accepted for publication once it meets all outstanding technical requirements.

Kind regards,

Yusuke Hoshino

Academic Editor

PLOS One
---

## [Editor Report · Acceptance letter]

PONE-D-25-54861R1

PLOS One

Dear Dr. Kidane,

I'm pleased to inform you that your manuscript has been deemed suitable for publication in PLOS One. Congratulations! Your manuscript is now being handed over to our production team.

Kind regards,

on behalf of

Dr. Yusuke Hoshino

Academic Editor

PLOS One